# The Epidemiological Transition of Surgically Treated Proximal Hip Fractures in Austria over the Course of the Pandemic—Back to Normal or a New Normal?

**DOI:** 10.3390/healthcare11243110

**Published:** 2023-12-07

**Authors:** Domenik Popp, Arastoo Nia, Sara Silvaieh, Thomas Sator, Thomas M. Tiefenboeck, Lukas Schmoelz, Rita Babeluk, Stefan Hajdu, Harald K. Widhalm

**Affiliations:** 1Clinical Division of Traumatology, Department of Orthopedics and Trauma Surgery, Medical University of Vienna, 1090 Vienna, Austriathomas.sator@meduniwien.ac.at (T.S.); thomas.tiefenboeck@meduniwien.ac.at (T.M.T.); lukas.schmoelz@meduniwien.ac.at (L.S.); stefan.hajdu@meduniwien.ac.at (S.H.); harald.widhalm@meduniwien.ac.at (H.K.W.); 2Department of Neurology, Medical University of Vienna, 1090 Vienna, Austria; sara.silvaieh@meduniwien.ac.at

**Keywords:** COVID, mortality, new normal, proximal hip fracture

## Abstract

Background: The COVID-19 pandemic has had a significant impact on the treatment protocols of orthopedic and trauma departments, but its specific effect on the mortality of hip fracture patients due to possible delays in surgery remains uncertain. This study aimed to investigate whether the COVID-19 pandemic worsened the mortality of patients with hip fractures. Materials and methods: This study included 246 prospectively enrolled patients who suffered from hip fractures during the Austrian State of Emergency period between 1 March and 30 June 2020 and 2021 and were admitted to a tertiary care trauma center. This cohort was compared with a retrospective control group of 494 patients admitted for hip fractures during the same timeframe in 2017, 2018, and 2019. These groups were compared to a prospective recruited “post-COVID-19 collective consisting of the years 2022 and 2023 including 313 patients. Results: This study found a 22% reduction in admissions during the COVID-19 period compared to the pre-COVID period (*p* = 0.018), as well as significant changes in gender (*p* = 0.013) and place of accident (*p* = 0.049). No other changes in demographic variables were observed. The 30-day mortality rate was 14.67% in the pre-COVID period, compared to 15.18% during the COVID-19 period (*p* = 0.381). No differences were observed in surgical complication rates or in the relationship between comorbidity burden and survival. Conclusion: This study did not show a higher perioperative mortality rate due to COVID-19. However, under current circumstances, with potentially reduced surgical and hospital bed capacities, it is expected that this condition might require a high degree of resources in times when resources are potentially scarce, such as during an ongoing pandemic. Level of evidence: Level III.

## 1. Background

The impact of COVID-19 on healthcare systems worldwide has been a major focus of many investigations. The exponential spread of COVID-19 has presented significant challenges for healthcare systems globally, altering the provision of healthcare and necessitating the reallocation of resources and staff to address shortages in personal protective equipment, disinfectants, and personnel on sick leave or under quarantine [1,2]. Nevertheless, the long-term effects of the pandemic on orthopedic and trauma surgery departments have not yet been sufficiently examined.

While the initial impact of COVID-19 on the Austrian healthcare system was moderate, orthopedic and trauma departments faced substantial challenges related to surgical and hospital bed capacities. Patients requiring urgent surgical intervention had to wait for Polymerase Chain Reaction (PCR) test results, leading to delays in treatment. Non-deferrable surgeries were performed under strict hygiene protocols without PCR test results [1]. The resulting reduction in staff and unpredictable trauma events have placed increased pressure on healthcare providers, requiring a delicate balance between staff protection and patient care. Although major and activity-related trauma cases have decreased, fragility fractures remain unchanged [2,3,4].

The pandemic has had a disproportionate impact on the elderly population, who are at higher risk of severe COVID-19 progression [5,6,7] and potential delays in operative treatment that may worsen outcomes [7,8,9]. Although only a small case series is available as a scientific source [10], the impact of COVID-19 on hip fracture patients and its effects on mortality and morbidity require further investigation.

Fractures occurring in the proximal femur can be transformative experiences for elderly individuals and are frequently linked to a significantly higher risk of mortality. The management of these fractures involves a standardized diagnostic and therapeutic algorithm to minimize complications and optimize resources [11,12].

Our hospital follows the national guidelines for perioperative assessment, with the aim of preparing and treating patients within 48 h of arrival at the emergency department, except for those ineligible for surgery. Surgical delays are associated with increased mortality and short-term complications [13]. The pandemic led to a reduction in traumatic emergencies, whereas the number of proximal femoral fractures remained the same. Elective knee and hip arthroplasties were almost completely postponed [14]. Compliance with the national and WHO guidelines resulted in the suspension of elective procedures, necessitating a swift strategy for safe trauma patient management [15].

During the COVID-19 lockdown, as elderly individuals limited their activities and stayed indoors, a decrease in hip fracture cases was anticipated. The expanded use of operating theaters for intensive care overflow, alongside challenges in accessing timely surgery, adds complexity to hip fracture care in Austria, raising interest in how COVID-19 affected hip fracture rates and treatment [16].

Therefore, our study aimed to investigate the changes in the epidemiology of hip fracture patients admitted to a level I trauma unit during the course of the COVID-19 pandemic, their survival rates, and whether there was a lasting effect on this condition.

## 2. Materials and Methods

This study was conducted in the Department of Orthopedics and Trauma Surgery at the Medical University of Vienna. This was a prospective cohort study with a retrospective control group carried out over seven different periods: the 120-day Austrian State of Emergency in 2020 (1 March to 30 June); the same timeframe in 2021, referred to as COVID; and the same period in 2017, 2018, and 2019, labeled as pre-COVID. To unmask a trend for the same period, 2022 and 2023 were evaluated and summarized as post-COVID. The prospectively collected COVID and post-COVID groups were compared with the retrospective pre-COVID cohort. The study was conducted in accordance with the STROBE guidelines for observational studies [17]. Hip fractures are common injuries among the elderly population, and the COVID-19 pandemic has had a significant impact on healthcare systems worldwide. This study aimed to investigate potential differences in the incidence, demographics, and outcomes of hip fractures during the COVID-19 pandemic compared to the pre-COVID period. The inclusion criteria were carefully selected to ensure a homogeneous study population, whereas the retrospective control group provided a baseline for comparison. The inclusion criteria encompassed individuals with a proximal femur fracture, aged 18 years or older, and who underwent treatment within the specified time frames. Utilizing the AO/OTA classification for adult cases, proximal femoral fractures (AO-31) were categorized into medial (AO-31B+C), pertrochanteric (AO-31A.1 and AO-31A.2), and subtrochanteric (AO-31A.3) fractures [18]. Exclusion criteria comprised non-operatively treated hip fractures, individuals below 18 years of age, and periprosthetic fractures. Additionally, patients with fractures in the middle or distal parts of the femur, periprosthetic femoral fractures, fractures resulting from polytrauma, or pathological fractures were excluded [19].

Our clinic adheres to a standard preoperative protocol, managed by a multidisciplinary medical team comprising trauma surgeons, anesthesiologists, and internal medicine specialists. Intracapsular fractures, unless displaced, were subject to internal fixation, while displaced cases underwent arthroplasty. Extracapsular fractures, including inter- and subtrochanteric fractures, were treated through internal fixation using either short or long nails.

By examining multiple time periods, this study aimed to provide a comprehensive analysis of the impact of the COVID-19 pandemic on hip fractures. Data regarding mortality rate, as the primary outcome measure, and causes of death were obtained by linking patient data with the registry of deaths from the Austrian Federal Institute for Statistics.

The use of the STROBE guidelines ensured the transparency and reproducibility of the study results [17].

### Statistical Analysis

Descriptive statistics were initially calculated for the main parameters, including the categorical variables, such as absolute and relative frequencies, and continuous variables, such as the number of observations, mean, standard deviation (SD), median, minimum, and maximum, respectively, for each year. Categorical variables were analyzed using the chi-squared test. The Kolmogorov–Smirnov test was computed to test for a violation of the normal distribution of our continuous variables. Continuous variables were normally expressed as medians and ranges and compared using the Mann–Whitney U test.

To identify potential differences between the cohorts in patient characteristics, the chi-square test was used for place of accident, fracture type, surgical treatment, and ASA score, whereas the *t*-test/Mann–Whitney U test was applied for ordinal variables such as patient age. To compare hip care (defined as the duration between accident and surgery and type of surgery) between the COVID and pre-COVID periods, either a two-sided t-test (if assumptions were not violated) or a two-sided U-test was performed.

Postoperative complications included anemia (HB < 9 g/dl or the need for blood transfusions), infection (local inflammation, elevated CRP levels, fever, and no signs of pneumonia on X-rays), pneumonia, delirium, thrombosis, and cardiopulmonary effects. Infections and pneumonia were separated due to the pandemic.

Statistical significance was set at *p* < 0.05. Only the results of the Cox model with the main effects are presented in the case of insignificant interaction terms. Subsequently, multivariate Cox regression analysis was conducted using significant variables from the univariate regressions (age, hip care, and postoperative complications). All statistical analyses were performed using SPSS software for Mac (version 21, IBM, SPSS).

## 3. Results

In total, 1053 patients were screened for further investigation. The pre-COVID group consisted of 494 patients from 2017, 2018, and 2019, while 246 patients were included in the COVID period, with 120 hip fractures in 2020 and 126 in 2021, whereas the post-COVID group consisted of 313 participants (151 patients in 2022 and 162 in 2023). After applying the inclusion and exclusion criteria, 1002 participants were included in the study (Figure 1).

The mean age in 2017 was 80.13 (SD: 12.02; Range: 38–100), in 2018—79.89 (SD: 11.85; Range 46–101), in 2019—79.88 (SD: 11.95; Range 49–99), in 2020—80.15 (SD: 11.78; Range 39–106), in 2021—79.97 (SD: 11.82; Range 43–100), in 2022—80.08 (SD: 11.55; Range 48–99), and in 2023—79.95 (SD: 11.69; Range 52–101) (*p* = 0.23).

Cox regression models were used to determine the relationship between time period and mortality risk while controlling for other factors such as ASA grade, age, sex, fracture type, and time until surgery. Our univariate Cox analyses indicated that mortality risk was significantly associated with factors such as age over 80, male gender, delayed surgery for more than 48 h, and pertrochanteric fractures (Table 1). However, the multivariate Cox regression analysis revealed that age and gender were the primary dependent variables for mortality risk (Table 2).

### 3.1. Pre-COVID vs. COVID

A statistically significant reduction of 25% in admissions due to proximal hip fractures was observed during the COVID period (*p* = 0.011) compared to the pre-COVID period. After excluding patients who were unfit for surgery and those with severe comorbidities, polytrauma, and pathological fractures, 702 patients were included in further analyses for these periods (Table 1). The majority of patients were over 60 years old (94% pre-COVID vs. 96% COVID, *p* = 0.095), and there were no significant differences in age distribution between the two periods. The proportion of female patients was higher in the pre-COVID group compared to the COVID group (69.88% vs. 61.87%, *p* = 0.021). The percentages of hip fractures and hip fracture surgeries were similar across the two periods (Table 1). The mean age of patients in each year did not significantly differ between the pre-COVID and COVID periods.

No statistically significant differences were found in the postoperative complication rates between the pre-COVID and COVID groups (Table 3). Of the 702 patients, 188 (38.05%) in the pre-COVID group experienced 210 postoperative complications. Similarly, 94 patients (38.21%) experienced 98 complications during the COVID period (*p* = 0.19). The most common postoperative complication in both groups was anemia, which accounted for 70.53% of all complications in the pre-COVID group and 68.38% in the COVID group (*p* = 0.31).

The mortality rate within 30 days of surgery was 15.29% before COVID and 15.59% during the pandemic, with no significant difference being detected between the two periods.

Notably, there was a significant reduction in the mean percentage of outdoor injuries during the COVID period compared with the pre-COVID period (Table 3). Additionally, an increase in the number of fractures in male patients was observed during the pandemic. However, there were no significant differences in demographic data, age, BMI, smoking habits, or chronic underlying medical conditions such as previous myocardial infarction, diabetes, dementia, or COPD during both periods. The choice of discharge facility and number of spinal anesthesia procedures for surgery remained constant throughout both periods (Table 4).

### 3.2. Pre-COVID vs. 2020

During the COVID period in 2020, there was a delay in the time from admission to surgery for proximal hip fractures. The majority (67.26%) of patients underwent surgery within 24 h, and 84.07% underwent surgery within 48 h. This is in contrast to the pre-COVID group, in which 86.46% underwent surgery within 24 h (96.37% within 48 h). The median time between the accident and surgery was 6.5 h in the pre-COVID period and 21 h in 2020. COVID safety precautions, such as PCR testing, were responsible for a delay of approximately 24 h (*p* = 0.025).

There were almost no significant changes compared with the pre-COVID period in terms of most of the studied values related to proximal hip fractures, except for some slight changes. For example, the number of fractures (*p* = 0.014) and incidence of outdoor injuries (*p* = 0.035) decreased significantly, and the male sex appeared to be more commonly affected by proximal hip fractures (*p* = 0.015). However, there were no significant variations in demographic data, age, BMI, comorbidities, complications, ASA score, fracture type, surgery type, anesthesia type, or mortality (Table 4).

### 3.3. COVID: 2020 vs. 2021

In 2021, the time from admission to surgery for proximal hip fractures decreased significantly compared to 2020 and approached pre-COVID levels, with a higher percentage of patients undergoing surgery within 24 h (*p* = 0.013) and 48 h (*p* = 0.038). The median duration between the accident and surgery also significantly decreased from 21 h in 2020 to 8 h in 2021 (*p* = 0.041). Additionally, there was a trend towards pre-COVID figures for the place of accident, with a 75.00% increase in outdoor injuries being observed (*p* = 0.029).

### 3.4. Pre-COVID vs. 2021

It seems that in 2021, there were almost no significant changes in comparison to the Pre-COVID period in terms of most of the studied values related to proximal hip fractures, except for some slight changes. For example, the male gender still appeared to be more commonly affected by proximal hip fractures, and the admission-to-surgery time narrowed from 6.5 h in the pre-COVID period to 8 h in 2021. Additionally, there was still a decrease of approximately 66.13% in outdoor injuries in 2021 compared to the pre-COVID group, although this decrease was not as significant as that in 2020. These changes were statistically significant, with *p*-values of 0.044, 0.039, and 0.031, respectively.

### 3.5. COVID vs. Post-COVID

A statistically significant increase of 27.23% in admissions due to proximal hip fractures was observed for the post-COVID period (*p* = 0.038) compared with the COVID group. After applying the exclusion criteria, 559 patients were included in further analyses for these periods (Table 1). The majority of patients were over 60 years of age (96% COVID vs. 95% post-COVID, *p* = 0.24), and there were no significant differences in age distribution between the two periods. The proportion of female patients was higher in the post-COVID group than in the COVID group (69.33% vs. 61.87%, *p* = 0.041). In general, the percentages of hip fractures and hip fracture surgeries were similar between the two periods (Table 1). The mean age of the patients in each year did not differ significantly between the post-COVID and COVID periods.

No statistically significant differences were found in postoperative complication rates between the post-COVID and COVID groups (Table 3). Of the 313 patients, 120 (38.33%) in the post-COVID group experienced 130 postoperative complications. The most common postoperative complication in both groups was anemia, which accounted for 70.74% of all complications in the post-COVID group and 68.38% in the COVID group (*p* = 0.28).

The mortality rate within 30 days of surgery was 15.99% after COVID and 15.59% during the pandemic, with no significant difference being detected between the two periods. 

A significant increase in the mean percentage of outdoor injuries was observed (Table 3). Additionally, a decrease in the number of fractures in male patients was observed in the post-COVID period. However, there were no significant differences in demographic data, age, BMI, or medical conditions between the two periods. The choice of discharge facility and number of spinal anesthesia procedures for surgery remained constant throughout both periods (Table 5).

During the post-COVID timeframe, a statistically significant increase of 27.23% in hospital admissions attributable to proximal hip fractures was observed compared with the COVID period (*p* = 0.038). Following the application of the exclusion criteria, the analysis focused on 559 patients for these durations (refer to Table 1 for details).

The majority of patients, accounting for 96% during COVID and 95% during post-COVID, were aged over 60 years, and there were no notable variations in age distribution between these periods (*p* = 0.24). However, the percentage of female patients was higher in the post-COVID group than in the COVID group (69.33% vs. 61.87%, *p* = 0.041). In general, the occurrence of hip fractures and the frequency of hip fracture surgeries were comparable between the two periods, as presented in Table 1. Furthermore, the mean patient age for each year did not exhibit significant divergence between the post-COVID and COVID periods.

Upon comparing the post-COVID and COVID groups (refer to Table 2), no statistically significant differences emerged in the incidence of postoperative complications. Among the 313 patients, 120 (38.33%) in the post-COVID group had 130 postoperative complications. Notably, the prevailing postoperative complication in both groups was anemia, constituting 70.74% of all complications in the post-COVID group and 68.38% in the COVID group (*p* = 0.28).

The 30-day post-surgery mortality rate was 15.99% post-COVID and 15.59% during the pandemic, with no significant disparity between these periods being identified. It is worth highlighting that there was a significant elevation in the mean percentage of outdoor injuries (refer to Table 2). Additionally, a reduction in the number of fractures in male patients was noted during the post-COVID period. However, there were no significant variations in demographic data, age, BMI, or prevailing medical conditions between the two periods. Consistency was observed in the selection of discharge facility and the utilization of spinal anesthesia for surgery throughout the entire period (as depicted in Table 5).

### 3.6. Pre-COVID vs. Post-COVID

It seems that for the pre-COVID and post-COVID periods, there were no significant changes in terms of most of the studied values related to proximal hip fractures.

It appears that male sex was equally affected by proximal hip fractures in these periods (*p* = 0.81), and the admission-to-surgery time reached the pre-COVID benchmark of 6.5 h in the post-COVID group (*p* = 0.53). Additionally, there was no significant difference in the number of outdoor injuries between the two groups (*p* = 0.69).

## 4. Discussion

The COVID-19 pandemic has had a significant impact on the healthcare industry, especially in orthopedic and trauma departments where acute surgical procedures are performed. This pandemic has caused several complications in acute surgical settings, leading to changes in patient management and healthcare resources. Unfortunately, studies on the outcomes of patients with hip fractures during the pandemic is limited, which has led to the need to evaluate the effects of the pandemic on treatment outcomes.

To address this need, the authors of this study aimed to provide data on the demographic and management changes that occurred during the pandemic and their impact on the short-term mortality of patients with hip fractures. Their findings indicate that there was a one-day delay in surgery during the first year of the pandemic, likely due to the challenges and restrictions imposed by the pandemic. However, in 2021, admission to surgery narrowed to almost pre-COVID levels, likely because of the widespread availability of faster and newer PCR test kits, which reduced the time for a COVID result from 20 to 70 min. This trend was supported, as the post-COVID period showed no differences from the pre-COVID period, hence the statement “back to normal”.

According to Nia et al. [20], there was a significant decrease of almost 70% in all trauma patient admissions by 2020. However, the incidence of hip fractures was only minimally affected by the social lockdown during the COVID period, with only a 22% decrease being observed. This resulted in a significant increase in the percentage of hip fracture surgeries compared with all surgeries during COVID, showing milder changes in hip fracture incidence during the pandemic than in other trauma cases [21].

This study also found that short-term mortality did not increase in patients with hip fractures during the emergency period. This suggests that the lower incidence of hip fractures during this period was likely associated with the State of Emergency and the stay-at-home strategy rather than the higher mortality due to COVID-19. Other studies have also reported a reduction in admissions for hip fractures, which may be attributed to changes in social behavior and related physical activity patterns during the lockdown, as a decrease in outdoor-related hip fractures was observed in this study during COVID [22]. Staying at home during the lockdown and avoiding outdoor activities under various weather conditions may have contributed to the reduction in hip fracture numbers, as many falls occur during normal daily activities.

While fear of infection in outpatient departments may explain the reduction in admissions for most nonsurgical diseases, such as kidney failure or myocardial infarction, this explanation is not applicable to hip fractures, as surgical treatment is usually inevitable and cannot be delayed. Another possible explanation for the comparatively milder reduction in hip fractures is the patient collective, as over 90% of the patients in all groups were older than 60 years with severe comorbidities such as osteoporosis, malnutrition, and sarcopenia (Table 3).

However, limited information is available on hip fractures and their survival rates during the global pandemic. Similar to Hershkowitz et al. [23] and other studies [24], no associations were found between comorbidity burden and patient survival, potentially due to the homogeneity of the patient group and the common prevalence of cardiovascular diseases, as well as reduced nutritional and physical status [23,25,26,27].

In this study, perioperative mortality remained unchanged and was consistent with national statistics and other published data [2,3,4,28]. Our multivariate Cox Regression analysis identified old age and male sex as significant factors for mortality [22].

Owing to the impact of COVID-19, several changes must be implemented, such as reduced operating room capacity and staffing. This information can be valuable for other medical professionals when making decisions. During a pandemic, hip fractures may require a greater percentage of resources because of patient needs that are not reduced.

The media often talk about a new normal. However, in the case of COVID and proximal hip fractures, the data from the post-COVID group showed a fast return to old values before the pandemic. Differences between these timeframes were rare and showed a relatively rapid return to pre-COVID levels. Therefore, in the case of proximal hip fractures, it is safer to return to normal rather than to a new normal.

This study is not without limitations. The retrospective nature of one of the control groups may have introduced bias in the measurement of laboratory parameters and history of disease compared with the prospective protocol for the COVID and post-COVID periods. However, the demographic data retrospectively collected in 2017, 2018, and 2019 and prospectively at hospital admission from 2020 to 2023 are unlikely to impact treatment decisions because of the short period between admission and surgery and the mandatory surgical indications in most cases.

## 5. Conclusions

In summary, this study found that patients with hip fractures during the COVID-19 pandemic had similar comorbidity and demographic profiles to those in the pre-COVID and post-COVID periods and did not have significantly increased short-term mortality rates. However, this study highlights the potential strain that hip fractures may cause on healthcare resources during a pandemic, given their prevalence and the need for surgical intervention. A comparison of the pre-COVID and COVID groups with the post-COVID group showed that the pandemic had little influence on hip fractures.

## Figures and Tables

**Figure 1 healthcare-11-03110-f001:**
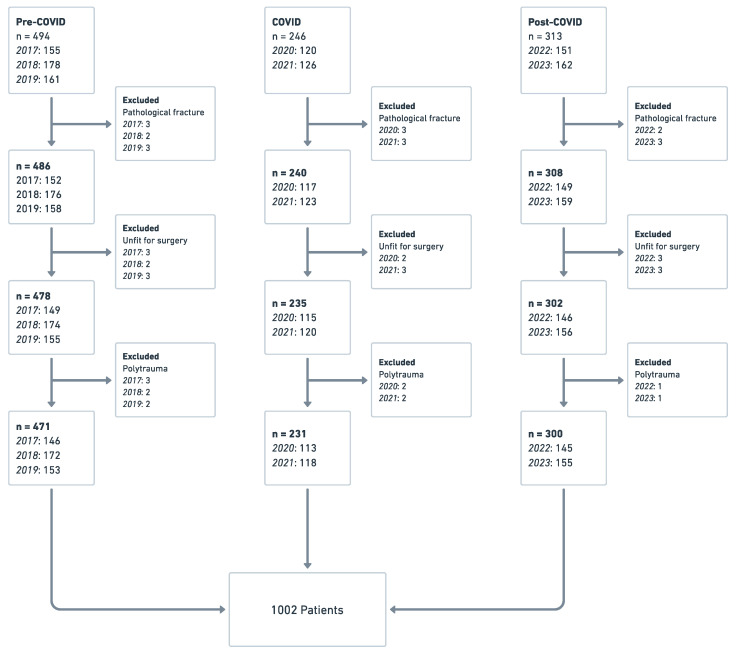
Flowchart of the overall study population.

**Table 1 healthcare-11-03110-t001:** Epidemiological data—surgical and fracture characteristics.

	2017		2018		2019		2020		2021		2022		2023	
	n	%	n	%	n	%	n	%	n	%	n	%	n	%
**Total**	155		178		161		120		126		151		162	
**Excluded**	9		6		8		7		8		6		7	
**Patho Fx**	3	33.33	2	33.33	3	37.50	3	42.80	3	37.50	2	33.33	3	42.86
**Unfit**	3	33.33	2	33.33	3	37.50	2	28.60	3	37.50	3	50.00	3	42.86
**Polytrauma**	3	33.33	2	33.33	2	25.00	2	28.60	2	25.00	1	16.67	1	14.28
***p*-value**	0.66		0.78		0.51		-		0.34		0.59		0.76	
**Gender**	146		172		153		113		118		145		155	
**male**	44	30.14	53	30.81	45	29.41	45	39.82	43	36.44	45	31.03	47	30.32
**female**	102	69.86	119	69.19	108	70.59	68	60.18	75	63.56	100	68.97	108	69.68
***p*-Value**	0.024		0.013		0.008		-		0.42		0.021		0.033	
**BMI**	23.66	(4.1)	23.88	(3.8)	23.72	(4.2)	22.98	(3.9)	23.13	(4.2)	23.53	(3.5)	23.79	(3.8)
***p*-Value**	0.65		0.59		0.74		-		0.88		0.83		0.92	
**Surgery**	146	94.19	172	96.63	153	95.03	113	94.67	118	93.65	145	96.02	155	95.68
***p*-Value**	0.021		0.008		0.013		-		0.10		0.042		0.028	
**Fx type**														
**Intra. n.d.**	12	7.53	14	8.15	12	7.84	8	7.08	9	7.62	12	8.28	13	8.39
**Intra**	58	39.73	64	37.20	60	39.22	43	38.05	44	37.29	57	39.31	59	38.06
**Petrochanteric**	67	45.89	82	47.67	71	46.40	55	48.68	57	48.31	65	44.83	70	45.16
**Subtrochanteric**	10	6.85	12	6.98	10	6.54	7	6.19	8	6.78	11	7.58	13	8.39
***p*-Value**	0.92		0.24		0.09		-		0.71		0.12		0.64	
**Surgery type**														
**HEP**	43	29.45	48	27.90	46	30.07	32	28.32	33	27.97	42	28.97	45	29.03
**THA**	15	10.27	16	9.31	14	9.16	11	9.73	11	9.32	15	10.34	14	9.03
**IM Nail**	65	44.52	79	45.93	69	45.09	52	46.02	55	46.61	65	44.83	71	45.81
**DHS**	11	7.53	15	8.72	12	7.84	10	8.85	10	8.47	11	7.58	12	7.74
**P** **.C.S.**	12	8.23	14	8.14	12	7.84	8	7.08	9	7.63	12	8.28	13	8.39
***p*-Value**	0.13		0.25		0.43		-		0.09		0.28		0.10	

Patho Fx: pathological fracture; intra. n.d: intracapsular not dislocated; THA: total hip arthroplasty; HEP: hemiarthroplasty; IM: intramedullary; DHS dynamic hip screw; P.C.S percutaneous cannulated screw.

**Table 2 healthcare-11-03110-t002:** Multivariate cox regression for OS.

	HR	CI Lower Bound	CI Upper Bound	*p*-Value
Age	1.079	1.023	1.231	0.041
Dur ad op	1.444	0.889	2.164	0.24
Pertroch Fx	1.459	0.978	2.019	0.24
Gender	1.071	1.025	1.213	0.029

Dur ad op: duration between admission and surgery; Pertroch Fx: pertrochanteric. Note: HR—Hazard ratio; CI—confidence interval.

**Table 3 healthcare-11-03110-t003:** Epidemiological data—surgical and general characteristics.

	2017		2018		2019		2020		2021		2022		2023	
	n	%	n	%	n	%	n	%	n	%	n	%	n	%
**Time to surgery**														
**<24 h**	127	86.99	147	85.47	133	86.93	76	67.26	97	82.21	124	85.52	133	85.81
**<48 h**	139	95.21	166	96.51	149	97.39	95	84.07	112	94.92	139	95.86	149	96.13
***p*-Value**	0.009		0.018		0.034		-		0.027		0.011		0.028	
**ASA**														
**1**	10	6.85	9	5.23	6	3.92	6	5.31	7	5.93	8	5.52	7	4.52
**2**	48	32.88	58	33.72	52	33.99	35	30.97	36	30.84	45	31.03	52	33.55
**3**	83	56.85	100	58.14	92	60.13	68	60.18	72	61.02	88	60.69	92	59.35
**4**	5	3.42	5	2.91	3	1.96	4	3.54	3	2.54	4	2.76	4	2.58
***p*-Value**	0.83		0.10		0.23		-		0.44		0.81		0.69	
**Place of accident**														
**Home**	103	70.55	124	72.09	113	73.86	104	92.04	103	87.29	105	72.42	111	71.62
**Outdoor**	41	28,08	45	26.16	38	24.84	8	7.08	14	11.86	39	26.89	42	27.09
**Hospital**	2	1.37	3	1.75	2	1.30	1	0.88	1	0.85	1	0.69	2	1.29
***p*-Value**	0.025		0.041		0.039		-		0.016		0.023		0.035	
**Complications**	68		76		66		48		50		63		67	
**Anemia**	47	69.12	53	69.74	48	72.73	33	68.75	33	66.00	44	69.84	48	71.64
**Infection**	3	4.41	3	3.95	2	3.03	2	4.17	2	4.00	3	4.76	2	2.99
**Delirium**	5	7.35	6	7.89	4	6.06	3	6.25	4	8.00	4	6.36	4	5.97
**Card/pulm aff**	4	5.88	5	6.58	3	4.54	3	6.25	3	6.00	3	4.76	4	5.97
**Thrombosis**	3	4.42	3	3.95	2	3.03	2	4.17	3	6.00	3	4.76	4	5.97
**Pneumonia**	6	8.82	6	7.89	7	10.61	5	10.41	5	10.00	6	9.52	5	7.46
***p*-Value**	0.43		0.81		0.10		-		0.66		0.58		0.73	
**Mortality**	23	15.75	26	15.11	23	15.03	18	15.93	18	15.25	23	15.86	24	16.13
***p*-Value**	0.66		0.55		0.49		-		0.69		0.75		0.88	

ASA: American association of Anesthesiology; Card/pulm aff: Cardiopulmonary affections.

**Table 4 healthcare-11-03110-t004:** Epidemiological and demographic characteristics.

	2017		2018		2019		2020		2021		2022		2023	
	n	%	n	%	n	%	n	%	n	%	n	%	n	%
**Tobacco Smoker**														
**Current**	5	3.43	6	3.49	5	3.27	4	3.54	4	3.39	5	3.45	6	3.87
**Former**	36	24.66	43	25.00	39	25.49	29	25.66	31	26.27	36	24.83	41	26.45
**Never**	90	61.64	105	61.05	93	60.78	68	60.18	72	61.02	90	62.07	94	60.65
**Unknown**	15	10.27	18	10.46	16	10.46	12	10.62	11	9.32	14	9.65	14	9.03
***p*-Value**	0.68		0.85		0.66		-		0.44		0.73			
**Hx of MCI**	11	7.53	14	8.14	12	7.84	8	7.08	8	6.78	10	6.89	12	7.74
***p*-Value**	0.71		0.81		0.58		-		0.68		0.52		0.10	
**Cong Heart Fail**	9	6.16	9	5.23	10	6.54	7	6.19	7	5.93	9	6.21	10	6.45
***p*-Value**	0.33		0.94		0.35		-		0.94		0.67		0.19	
**PVD**	8	5.48	12	6.98	9	5.88	6	5.31	7	5.93	8	5.52	9	5.81
***p*-Value**	0.65		0.75		0.45		-		0.087		0.29		0.56	
**COPD**	10	6.85	13	8.90	13	8.50	8	7.08	9	7.63	11	7.59	12	7.74
***p*-Value**	0.43		0.77		0.69		-		0.48		0.81		0.24	
**Dementia**	25	17.12	29	16.86	27	17.65	19	16.81	20	16.95	25	17.24	26	16.77
***p*-Value**	0.58		0.61		0.07		-		0.26		0.09		0.74	
**Hx of Stroke**	5	3.42	8	4.65	5	3.27	4	3.54	6	5.08	7	4.83	7	4.52
***p*-Value**	0.14		0.91		0.12		-		0.85		0.15		0.93	
**DMII**	16	10.96	20	11.63	19	12.42	14	12.39	13	11.02	17	11.72	18	11.61
***p*-Value**	0.32		0.18		0.66		-		0.59		0.48		0.39	
**CKD**	9	6.16	10	5.81	10	6.54	6	5.31	6	5.08	9	6.21	10	6.45
***p*-Value**	0.28		0.06		0.10		-		0.27		0.81		0.07	
**Liver disease**	4	2.74	3	1.74	2	1.31	2	1.77	3	2.54	2	1.38	2	1.29
***p*-Value**	0.33		0.64		0.10		-		0.81		0.07		0.08	
**Discharge Facility**														
**Home**	68	46.90	79	45.93	69	45.10	51	45.14	53	44.92	66	45.52	69	44.52
**Retirement Home**	58	40.00	71	41.28	61	39.87	47	41.59	47	39.83	58	40.00	64	41.29
**P.T.C.**	15	10.34	19	11.05	17	11.11	12	10.62	13	11.02	15	10.34	18	11.61
**Hospice**	4	2.76	3	1.74	6	3.92	3	2.65	5	4.23	6	4.14	4	2.58
***p*-Value**	0.67		0.78		0.92		-		0.89		0.23		0.57	
**Anesthesia Type**														
**Spinal**	101	69.18	116	67.44	105	68.63	78	69.03	81	68.64	101	69.66	109	70.32
**General**	45	30l.82	56	32.56	48	31.37	35	30.97	37	31.36	44	30.34	46	29.68
***p*-Value**	0.11		0.39		0.69		-		0.37		0.23		0.56	

Hx: History; MCI: Myocardial Infarction; BMI: Body Mass Index; Cong Heart Fail: Congestive Heart Failure; PVD: Peripheral Vascular Disease; COPD: Chronic Obstructive Pulmonary Disease; DMII: Diabetes Mellitus Type 2; CKD: Chronic Kidney Disease; P.T.C.: Physical Therapy Center. Categorical variables are reported in n (% of total); continuous parametric variables are reported as mean (SD).

**Table 5 healthcare-11-03110-t005:** Cox univariate regression: overall survival is a dependent variable.

Risk Score	HR	(95%-CI)	*p*-Value
Age	1.069	(1.013–1.176)	0.021
Period	0.401	(0.084–1.979)	0.40
Dur ad op	1.599	(1.221–1.889)	0.009
Pertroch Fx	1.603	(1.144–1.895)	0.018
Gender	1.453	(1.123–1246)	0.006
ASA	0.502	(0.115–1.926)	0.07

Dur ad op: duration between admission and surgery; Pertroch Fx: pertrochanteric Fracture; ASA: American association of Anesthesiology. Note: HR—Hazard ratio; CI—confidence interval.

## Data Availability

Data are contained within the article.

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
