# Peer review of "The Epidemiological Transition of Surgically Treated Proximal Hip Fractures in Austria over the Course of the Pandemic—Back to Normal or a New Normal?"

_healthcare, 2023, doi:10.3390/healthcare11243110_

Round 1
Reviewer 1 Report
Comments and Suggestions for Authors
The article appears to be original, and the topic is interesting, especially in understanding the impact of the emergence of COVID on Austria's healthcare system and its effect on the described pathology. However, upon reviewing the paper, there are certain aspects that the authors should address for improvement.
Firstly, the title is a bit too concise and lacks sufficient information regarding the selected population and the specific area where the investigation was conducted.
I found the background paragraph to be somewhat sparse in addressing the researchers' questions. It would benefit from a more comprehensive approach, clarifying why this study is being conducted. Are there expected epidemiological changes in the number of fractures due to lifestyle changes, or is there an anticipated shift in the prognosis of fractures due to an overburdening of the national healthcare system?
The background section should include the study objective and be relocated from the "Materials and Methods" section. Additionally, the inclusion/exclusion criteria are unclear, and the population is described too briefly.
In Section 2.1, population characteristics are listed in Table 3 (it might be more logical to place them in Table 1), but some are not mentioned, and others are listed without corresponding information in this section. It's important to include references in the text.
Section 3.2 does not mention differences in where the incidents occurred, which is only addressed in Section 3.5 (a somewhat repetitive paragraph). Furthermore, there is a figure reported in the text regarding the data divided between males and females that is not shown in the tables.
While the overall execution of the work is commendable, there is a need for more order in the writing, including the numbering of tables, references, and explanations of acronyms. Additionally, a more extensive bibliography would enhance the completeness of the paper.
I suggest adding the following references:
Maranesi E, et al. Randomised controlled trial assessing the effect of a technology-assisted gait and balance training on mobility in older people after hip fracture: study protocol. BMJ Open. 2020 Jun 15;10(6):e035508. doi: 10.1136/bmjopen-2019-035508.
Bevilacqua R, et al. Rehabilitation of older people with Parkinson's disease: an innovative protocol for RCT study to evaluate the potential of robotic-based technologies. BMC Neurol. 2020 May 13;20(1):186. doi: 10.1186/s12883-020-01759-4.
Author Response
Thank you for the opportunity to revise following peer review.
We welcome the comments and suggestions for improvement provided. Our response to your comments on the manuscript are listed below.
- Firstly, the title is a bit too concise and lacks sufficient information regarding the selected population and the specific area where the investigation was conducted.
Thank you for your comment. We changed the title and tried to be more specific.
Epidemiological transition of surgically treated proximal hip fractures in Austria in the course of a pandemic – back to normal or a new normal?
- I found the background paragraph to be somewhat sparse in addressing the researchers' questions. It would benefit from a more comprehensive approach, clarifying why this study is being conducted. Are there expected epidemiological changes in the number of fractures due to lifestyle changes, or is there an anticipated shift in the prognosis of fractures due to an overburdening of the national healthcare system?
Thank you for your comment. We added a whole paragraph addressing proximal hip fractures, expected changes and we tried to clarify our motive for this study.
Fractures occurring in the proximal femur can be transformative experiences for elderly individuals, frequently linked to a significant higher risk of mortality. Managing these fractures involves a standardized diagnostic and therapeutic algorithm to minimize complications and optimize resources. Our hospital follows national guidelines for perioperative assessment, aiming to prepare and treat patients within 48 hours of arriving at the emergency department, except for those ineligible for surgery. Delays in surgery are linked to increased mortality and short-term complications. The pandemic led to a reduction in traumatic emergencies, whereas the number of proximal femoral fractures stayed the same. Elective knee and hip arthroplasty were almost completely postponed. Compliance with national and WHO guidelines resulted in the suspension of elective procedures, necessitating a swift strategy for safe trauma patient management. During the Covid-19 lockdown, as elderly individuals limited activities and stayed indoors, one might anticipate a decrease in hip fracture cases. The expanded use of operating theatres for intensive care overflow, alongside challenges in accessing timely surgery, adds complexity to hip fracture care in Austria, raising interest in how Covid-19 affected hip fracture rates and treatment.
- The background section should include the study objective and be relocated from the "Materials and Methods" section.
Thank you for your comment. We relocated the study objective to the end of the introduction part. The redundant part in the Materials and Methods section was deleted.
- Additionally, the inclusion/exclusion criteria are unclear, and the population is described too briefly.
Thank you for your comment. We changed and extended the Material and f d
Inclusion criteria were a proximal femur fracture, age ≥ 18 years and treatment received between the mentioned periods. Based on the AO/OTA classification for adults, proximal femur fractures (AO-31) were grouped into medial (AO-31B + C), pertrochanteric (AO-31A.1 and AO-31A.2) and subtrochanteric (AO-31A.3) fractures. Exclusion criteria were hip fracture treated non-operatively, age < 18 years or periprosthetic fractures. Patients presenting with fractures of the middle or distal third of the femur, periprosthetic femoral fractures, fractures due to polytrauma and pathological fractures were excluded.
- In Section 2.1, population characteristics are listed in Table 3 (it might be more logical to place them in Table 1), but some are not mentioned, and others are listed without corresponding information in this section. It's important to include references in the text.
Thank you for your comment. We rearranged the tables. Gender and BMI was shifted to Table 1. We hope that it is now in a reasonable order. Due to our approach to find changes during COVID-19 we assessed a lot of parameters, but most showed no significant changes. To focus on epidemiological shifts we do not mentioned all in the text. We wanted the tables to be an overview for readers interested in other details, and therefore mentioned the most interesting findings in the written part of the manuscript.
- Section 3.2 does not mention differences in where the incidents occurred, which is only addressed in Section 3.5 (a somewhat repetitive paragraph). Furthermore, there is a figure reported in the text regarding the data divided between males and females that is not shown in the tables.
Thank you for your comment.
Ad 3.2: Thank you for pointing this out. We added a whole paragraph and hope the results are now described acceptable.
There were almost no significant changes compared to the pre-COVID period in terms of most of the studied values related to proximal hip fractures, except for some slight changes. For example, the number of fractures (p=0.014) and incidence of outdoor injuries (p = 0.035) decreased significantly, and the male sex appeared to be more commonly affected by proximal hip fractures (p=0.015). However, there were no significant variations in demographic data, age, BMI, comorbidities, complications, ASA-Score, fracture type, surgery type, anesthesia type, or mortality observed.
Ad 3.5: We did not mean to differentiate between the genders in this part and rewrote the paragraph and tried to be more specific. Thank you for pointing out this inaccuracy.
In general the occurrence of hip fractures and the frequency of hip fracture surgeries were comparable between the two periods, as presented in Table 1. Furthermore, the mean patient age for each year did not exhibit significant divergence between the post-COVID and COVID periods.
- While the overall execution of the work is commendable, there is a need for more order in the writing, including the numbering of tables, references, and explanations of acronyms. Additionally, a more extensive bibliography would enhance the completeness of the paper.
Thank you for your comment. We re-organized and edited the whole manuscript. We hope the tables, references and acronyms are now fine. Thanks again for pointing out these mistakes.
- I suggest adding the following references:
Maranesi E, et al. Randomised controlled trial assessing the effect of a technology-assisted gait and balance training on mobility in older people after hip fracture: study protocol. BMJ Open. 2020 Jun 15;10(6):e035508. doi: 10.1136/bmjopen-2019-035508.
Bevilacqua R, et al. Rehabilitation of older people with Parkinson's disease: an innovative protocol for RCT study to evaluate the potential of robotic-based technologies. BMC Neurol. 2020 May 13;20(1):186. doi: 10.1186/s12883-020-01759-4.
Thank you for your comment and helpful literature suggestions. These papers were an excellent add on and we hope you appreciate the improvements. We hope that we addressed all of them to your satisfaction and look forward to your feedback.
Yours sincerely
Domenik Popp, MD, MSc
Consultant Orthopedics and Trauma Surgery
Medical University of Vienna
University Clinic of Orthopedics and Trauma Surgery
Waehringer Guertel 18-20, A-1090 Vienna, Austria
domenik.popp@meduniwien.ac.at
On behalf of all co-authors

Reviewer 2 Report
Comments and Suggestions for Authors
Dear Authors,
It is my pleasure to review your study but I have a lot of doubts.
General information:
-first of all, prepare the article for review in accordance with the journal's guidelines;
-in abstract line 13: "Patients and methods:" should be "Material and methods" - this part is unclear - please check it.
-in abstract line 15: "and were admitted to a level I trauma center." - what does it mean "level I trauma" ? It should be more clear.
Introduction:
-the introduction only addresses the issue of COVID-19, this part should be corrected, information about proximal hip fractures should be presented.
M&M:
- in line 62: "Patients diagnosed with a hip fracture (OTA/AO 31, 32.1) - it should be explained, for orthopedics it's clear but for the other specialist can be not,
-was the AO scale used or others as well?
-inclusion and exclusion criteria to the study should be presented;
-in line 75: "The use of the 75 STROBE guidelines..." please add reference
-lin line 101-110: this part should be moved, does not apply statistical analysis
Results:
-I have no objections.
Discussion:
- limitations section should be on the end of "Discussion" part
References:
-prepare the article in accordance with the journal's guidelines;
-please add the DOI address to each reference;
-references should be placed in square brackets;
Comments on the Quality of English LanguageEnglish language correction must be performed.
Author Response
Dear Reviewer 2,
Thank you for the opportunity to revise following peer review.
We welcome the comments and suggestions for improvement provided. Our responses to your comments on the manuscript are listed below.
General information:
1) -first of all, prepare the article for review in accordance with the journal's guidelines;
Thank you for your comment. We prepared the manuscript according to the journal's guidelines, e.g.: References are now in brackets, etc. Thank you for mentioning it.
2) -in abstract line 13: "Patients and methods:" should be "Material and methods" - this part is unclear - please check it.
Thank you for your comment. We made the required changes.
3) -in abstract line 15: "and were admitted to a level I trauma center." - what does it mean "level I trauma"? It should be more clear.
Thank you for your comment. We changed level I trauma center to the more commonly used tertiary care trauma center. Thank you for pointing this out.
Introduction:
4) -the introduction only addresses the issue of COVID-19, this part should be corrected, and information about proximal hip fractures should be presented.
Thank you for your comment. We added a whole paragraph addressing proximal hip fractures.
Fractures occurring in the proximal femur can be transformative experiences for elderly individuals, frequently linked to a significantly higher risk of mortality. Managing these fractures involves a standardized diagnostic and therapeutic algorithm to minimize complications and optimize resources. Our hospital follows national guidelines for perioperative assessment, aiming to prepare and treat patients within 48 hours of arriving at the emergency department, except for those ineligible for surgery. Delays in surgery are linked to increased mortality and short-term complications. The pandemic reduced traumatic emergencies, whereas the number of proximal femoral fractures stayed the same. Elective knee and hip arthroplasty were almost completely postponed. Compliance with national and WHO guidelines resulted in suspending elective procedures, necessitating a swift strategy for safe trauma patient management. During the Covid-19 lockdown, as elderly individuals limited activities and stayed indoors, one might anticipate a decrease in hip fracture cases. The expanded use of operating theatres for intensive care overflow, alongside challenges in accessing timely surgery, adds complexity to hip fracture care in Austria, raising interest in how Covid-19 affected hip fracture rates and treatment.
M&M:
5) - in line 62: "Patients diagnosed with a hip fracture (OTA/AO 31, 32.1)
- it should be explained, for orthopedics it's clear but for the other specialist can be not,
-was the AO scale used or others as well
-inclusion and hj
Thank you for your comment. We described the used classification in more detail and added more information about inclusion and exclusion criteria. I hope that it is now clear, not just for specialists.
Patients diagnosed with a hip fracture (OTA/AO 31, 32.1) during the COVID period were included in the study, while those with periprosthetic femoral fractures, fractures due to polytrauma, and pathological fractures were excluded.
Inclusion criteria were a proximal femur fracture, age ≥ 18 years, and treatment received between the mentioned periods. Based on the AO/OTA classification for adults, proximal femur fractures (AO-31) were grouped into medial (AO-31B + C), pertrochanteric (AO-31A.1 and AO-31A.2) and subtrochanteric (AO-31A.3) fractures. Exclusion criteria were hip fracture treated non-operatively, age < 18 years, or periprosthetic fractures. Patients presenting with fractures of the middle or distal third of the femur, periprosthetic femoral fractures, fractures due to polytrauma, and pathological fractures were excluded.
As a standard procedure, the preoperative setting in our clinic was performed by a multidisciplinary medical team consisting of trauma surgeons, anesthesiologists, and internal medicine specialists. Intracapsular fractures, if not displaced, were treated by internal fixation, whereas, if displaced, by arthroplasty. Extracapsular fractures such as inter- and subtrochanteric fractures were treated by internal fixation using short or long nails.
8) -in line 75: "The use of the 75 STROBE guidelines..." please add reference
Thank you for your comment. An appropriate reference was added.
Cuschieri S. The STROBE guidelines. Saudi J Anaesth. 2019 Apr;13(Suppl 1):S31-S34. doi: 10.4103/sja.SJA_543_18. PMID: 30930717; PMCID: PMC6398292.
9) -lin line 101-110: this part should be moved, does not apply statistical analysis
Thank you for your comment. The unnecessary part was deleted.
Delirium was defined as a change in the preoperative observed baseline mental functioning shortly after surgery. Cardiopulmonary affections included therapy-resistant systolic blood pressure > 160 mmHg, postoperative chest pain with abnormal ECG alterations, or dyspnea.
Discussion:
10) - limitations section should be on the end of "Discussion" part
Thank you for your comment. We deleted the “Limitations”-headline and put it at the end of the discussion section.
References:
11) -prepare the article in accordance with the journal's guidelines;
Thank you for your comment. We prepared the manuscript according to the journal's guidelines;
12) -please add the DOI address to each reference;
Thank you for your comment. DOI addresses were added.
13) -references should be placed in square brackets;
Thank you for your comment. The references are now in brackets.
Thank you for your comments. The manuscript is now prepared according to the journal's guidelines. We hope that we addressed all of them to your satisfaction and look forward to your feedback.
Yours sincerely
Domenik Popp, MD, MSc
Consultant Orthopedics and Trauma Surgery
Medical University of Vienna
University Clinic of Orthopedics and Trauma Surgery
Waehringer Guertel 18-20, A-1090 Vienna, Austria
domenik.popp@meduniwien.ac.at
On behalf of all co-authors

Reviewer 3 Report
Comments and Suggestions for Authors
The research aim is to investigate the effects of COVID-19 pandemic on the mortality rates of patients with hip fractures.
The abstract is written and structured appropriately.
The introduction transposes the research into the topic and formulates the objective of the study at the end. However, more specific information on how COVID-19 pandemic influenced Orthopedics and Traumatology should be provided in relation to other scientific papers, for e.g. Moldovan, F.; Gligor, A.; Moldovan, L.; Bataga, T. An Investigation for Future Practice of Elective Hip and Knee Arthroplasties during COVID-19 in Romania. Medicina 2023, 59, 314. doi: 10.3390/medicina59020314. Please rephrase the first sentence – line 32 (“As a scientific researcher”).
In the methodology section, the stages of the research are presented. In line 57 the hospital where the study was performed should be stated. In line 63 – it is stated the inclusion criteria as patients with hip fractures corresponding to AO classification 32 which in fact corresponds to diaphyseal fractures of the femur? The term “hip fractures” should be better defined here as there can be trochanteric, neck and head fractures or intracapsular/extracapsular hip fractures which is important from the point of view of the selected treatment. As concern for the statistical analysis, Kolmogorov–Smirnov is a test used in order to assess the normal distribution of variables; for normally distributed data you normally use the student T-test – please clarify.
Regarding results section, as a general rule of reporting the p values: if p value is greater than 0.05 should be reported with two decimal values, if p value is between 0.001 and 0.05 should be reported with three decimal places and if values shown on output as 0.000 should be reported as <0.0001; I suggest to make this correction throughout the paper because there is no consistency sometimes p-values are reported with 3 decimal places and sometimes with 2 decimal places without respecting any of the above mentioned rules.
The discussions interpret the research results and relate them to other results from scientific literature. Limitations of the study are provided.
The conclusions are concise and clear.
The references are adequate but need proper editing and can be extended as suggested above.
Author Response
Dear Reviewer 3,
Thank you for the opportunity to revise following peer review.
We welcome the comments and suggestions for improvement provided. Our responses to your comments on the manuscript are listed below.
- The introduction transposes the research into the topic and formulates the objective of the study at the end. However, more specific information on how COVID-19 pandemic influenced Orthopedics and Traumatology should be provided in relation to other scientific papers, for e.g. Moldovan, F.; Gligor, A.; Moldovan, L.; Bataga, T. An Investigation for Future Practice of Elective Hip and Knee Arthroplasties during COVID-19 in Romania. Medicina 2023, 59, 314. doi: 10.3390/medicina59020314.
Thank you for your comment and helpful literature suggestion. This paper was an excellent add on and we hope you are now satisfied with the introduction
- Please rephrase the first sentence – line 32 (“As a scientific researcher”).
Thank you for your comment. The sentence was re-phrased.
The impact of COVID-19 on healthcare systems worldwide has been a major focus of many investigations.
- In the methodology section, the stages of the research are presented. In line 57 the hospital where the study was performed should be stated.
Thank you for your comment. The hospital name was added.
- In line 63 – it is stated the inclusion criteria as patients with hip fractures corresponding to AO classification 32 which in fact corresponds to diaphyseal fractures of the femur? The term “hip fractures” should be better defined here as there can be trochanteric, neck and head fractures or intracapsular/extracapsular hip fractures which is important from the point of view of the selected treatment.
- As concern for the statistical analysis, Kolmogorov–Smirnov is a test used in order to assess the normal distribution of variables; for normally distributed data you normally use the student T-test – please clarify.
Thank you for your comment. The wording was misleading. We rewrote the paragraph and clarified the methods.
Kolmogorov–Smirnov test was computed to test for a violation of the normal distribution of our continuous variables. Continuous variables were normally expressed as median and range and compared with Mann- Whitney-U test.
- Regarding results section, as a general rule of reporting the p values: if p value is greater than 0.05 should be reported with two decimal values, if p value is between 0.001 and 0.05 should be reported with three decimal places and if values shown on output as 0.000 should be reported as <0.0001; I suggest to make this correction throughout the paper because there is no consistency sometimes p-values are reported with 3 decimal places and sometimes with 2 decimal places without respecting any of the above mentioned rules.
Thank you for your comment. We made the changes in the manuscript according to your suggestions.
- The references are adequate but need proper editing and can be extended as suggested above.
Thank you for your comment. The references are now edited to the journal's requirements, and new references were added. Thank you again for your literature recommendation. We hope that we addressed all of them to your satisfaction and look forward to your feedback.
Yours sincerely
Domenik Popp, MD, MSc
Consultant Orthopedics and Trauma Surgery
Medical University of Vienna
University Clinic of Orthopedics and Trauma Surgery
Waehringer Guertel 18-20, A-1090 Vienna, Austria
domenik.popp@meduniwien.ac.at
On behalf of all co-authors

Round 2
Reviewer 1 Report
Comments and Suggestions for Authors
The authors have answered all the suggesions I have made.
Author Response
Re: Manuscript ID: 2709688
Dear Reviewer 1,
thank you for your feedback and contribution to the significant improvement of this paper.
sincerely
Domenik Popp, MD, MSc
Consultant Orthopedics and Trauma Surgery
Medical University of Vienna
University Clinic of Orthopedics and Trauma Surgery
Waehringer Guertel 18-20, A-1090 Vienna, Austria
domenik.popp@meduniwien.ac.at
On behalf of all co-authors
Reviewer 2 Report
Comments and Suggestions for Authors
Dear Authors,
Thank you for proofreading the manuscript. It looks much better now. I suggest making a few more changes.
First: in line 96: "Based on the AO/OTA classification for adults, proximal femur fractures (AO-31) were grouped into medial (AO-31B + C)..." - please add references.
Second: I propose to present the inclusion and exclusion criteria starting from dashes/points.
Apart from that, I have no other objections.
Thank you very much for your cooperation.
Best regards,
Author Response
Re: Manuscript ID: 2709688
Dear Reviewer 1,
thank you for your feedback and contribution to the significant improvement of this paper.
Thank you for your comment. We added the following reference to clarify the classification issue so that readers unfamiliar with the topic can find an explanation for the AO scheme. Thank you for pointing that out.
Meinberg, E.; Agel, J.; Roberts, C.; Karam, M.; Kellam, J. Fracture and Dislocation Classification Compendium—2018. Journal of orthopaedic trauma 2018, 32, S1-S10, doi:10.1097/bot.0000000000001063.
regarding the inclusion/exclusion criteria: we hope you do not mind keeping it in the current format. As there are not many inclusion/exclusion criteria, we believe a format with dashes/points would occupy a lot of space and would not help make a more lucid manuscript. We hope you do not mind. If this is a major concern of yours, we will address it in a proper way of course.
Again, thank you for your excellent comments which helped to improve this manuscript significantly.
sincerely
Domenik Popp, MD, MSc
Consultant Orthopedics and Trauma Surgery
Medical University of Vienna
University Clinic of Orthopedics and Trauma Surgery
Waehringer Guertel 18-20, A-1090 Vienna, Austria
domenik.popp@meduniwien.ac.at
On behalf of all co-authors
Reviewer 3 Report
Comments and Suggestions for Authors
The authors have addressed all my concerns.
Author Response
Re: Manuscript ID: 2709688
Dear Reviewer 3,
thank you for your feedback and contribution to the significant improvement of this paper.
sincerely
Domenik Popp, MD, MSc
Consultant Orthopedics and Trauma Surgery
Medical University of Vienna
University Clinic of Orthopedics and Trauma Surgery
Waehringer Guertel 18-20, A-1090 Vienna, Austria
domenik.popp@meduniwien.ac.at
On behalf of all co-authors